# The Use of Nanocellulose in Edible Coatings for the Preservation of Perishable Fruits and Vegetables

Annachiara Pirozzi [1] , Giovanna Ferrari [1,2] and Francesco Donsì [1,*]

1 Department of Industrial Engineering, University of Salerno, Via Giovanni Paolo II, 132, 84084 Fisciano, Italy; apirozzi@unisa.it (A.P.); gferrari@unisa.it (G.F.)
2 ProdAl Scarl, University of Salerno, Via Giovanni Paolo II, 132, 84084 Fisciano, Italy
* Correspondence: fdonsi@unisa.it

**Abstract:** The usage of edible coatings (ECs) represents an emerging approach for extending the shelf life of highly perishable foods, such as fresh and fresh-cut fruits and vegetables. This review addresses, in particular, the use of reinforcing agents in film-forming solutions to tailor the physicochemical, mechanical and antimicrobial properties of composite coatings. In this scenario, this review summarizes the available data on the various forms of nanocellulose (NC) typically used in ECs, focusing on the impact of their origin and chemical or physical treatments on their structural properties (morphology and shape, dimension and crystallinity) and their functionality. Moreover, this review also describes the deposition techniques of composite ECs, with details on the food engineering principles in the application methods and formulation optimization. The critical analysis of the recent advances in NC-based ECs contributes to a better understanding of the impact of the incorporation of complex nanoparticles in polymeric matrices on the enhancement of coating properties, as well as on the increase of shelf life and the quality of fruits and vegetables.

**Keywords:** edible coating; shelf life; quality; barrier; cellulose; nanocomposite; reinforcing agent; nanofiller

## 1. Introduction

Traditional commercial food packaging materials, such as glass, aluminum, tin, and petroleum-based polymers, are widely used for the protection of goods from physical damage, external contamination or deterioration [1]. To limit the environmental pollution caused by non-degradable plastic packaging, the use of biocompatible macromolecules seems a promising strategy for a more sustainable packaging. In this frame, edible coatings (ECs) represent a consolidated technology to improve the postharvest quality of fruits and vegetables by slowing down respiration rate, water loss and oxidation processes [2], as well as helping to maintain the physiological properties.

ECs consist of a thin layers of proteins, polysaccharides or lipids, and are applied directly to the surface of the food in a liquid form with different techniques, forming a micro-layer film on the surface of the food [3]. ECs acts as primary (closest to food) packaging. Thus, the main advantage over traditional synthetic packaging is that ECs can be consumed with the food, with no package to dispose of [4], reducing the cost and complexity of packaging systems designed to protect fresh perishable foods. Moreover, also if they are not eaten by the consumers, ECs could still contribute to the reduction of environmental pollution, because they are produced exclusively from renewable, edible ingredients and therefore degrade more readily than polymeric materials [5]. In addition, because of the additional protection they offer, ECs enable also the simplification of the secondary packaging (next layer of packaging), making recycling more accessible [6]. However, the performance of most ECs is insufficient to meet practical applications, in particular with reference to conjugating restricted thickness with adequate mechanical and barrier properties. Therefore, recent studies have focused on the implementation of

different types of filler to improve coating properties [7]. The preservation action of ECs can be enhanced through the incorporation of a wide range of bioactive compounds, like aroma compounds, essential oils, antioxidants, pigments and ions [8–10], which contribute to slowing down the browning reactions of fresh and fresh-cut fruits and vegetables, as well as decreasing microbial growth, thereby leading to the shelf-life extension of the products [11]. Moreover, the synergistic interaction between reinforcement agents and the polymeric material, through hydrogen bonding or ionic complexation, enables the ECs mechanical properties to be increased and permeability of moisture and gases to be reduced [12].

Among the different reinforcing agents, nanofillers or additives with their size lying in the range of 100 nm have attracted increasing attention. In this scenario, nanocellulose (NC) emerged as a promising material for tailoring ECs properties in food preservation. For their nano-reinforcing effect in many different polymer matrices, three types of cellulose are mainly used, such as cellulose nanocrystals (CNC), cellulose nanofibrils (CNF), and bacterial nanocellulose (BNC).

This review describes the engineering and applicative aspects related to the use of NC in composite ECs to prevent the quality decay of perishable fruits and vegetables during storage. In this regard, after a brief overview of ECs deposition techniques and the optimization of their formulation, this review illustrates the property enhancement of ECs reinforced with NC and other active compounds as well.

## 2. Edible Coating Deposition and Optimization

### 2.1. Methods of Coating Application to Food Products

ECs can be applied to food products using different techniques (Figure 1), which are described below.

The selection of the most suitable deposition technique of the EC layer is generally based on (a) the characteristics of the foods to be coated, and in particular surface hydrophobicity and roughness, (b) the physical properties of the coating, such as viscosity, density and surface tension [13], as well as (c) the intended effect of the coating, (d) the available drying technique, (e) the intended industrial application, and (f) the cost.

#### 2.1.1. Spraying Method

Spraying (Figure 1a) is a conventional method for applying low-viscosity coating solutions on food surfaces, and is generally based on high-pressure atomization in the 60–80 psi (4.1–5.5 bar) range to produce fine droplets, which are deposited on the food products until a homogeneous and uniform layer is formed [14]. The thickness of the coating layer can be controlled through the atomization conditions and is generally associated with the hydrodynamic diameter of the droplet. Good atomization conditions are associated to the size distribution of the sprayed droplets, which depends on the main operating parameters, including the atomizer features (spray gun type, operating pressure, and nozzle temperature), and the main operating conditions, such as air and liquid flow rate, the humidity of incoming air and the polymer solution [15]. The volumes of coating solution required per unit mass of product to be coated are lower for spraying than for other processes. For all these reasons, spraying is especially recommended when a high-quality product is desired for large-scale productions. However, the thickness of the coating layer that can be obtained by spraying is limited by the lower limit of the hydrodynamic diameter of the droplets that can be obtained by atomization, which is of around 20 μm, whereas in electrospraying processes (discussed in the following section), an aerosol with droplets below 100 nm can be generated [13,16,17]. The final coating quality in spraying processes depends also on the drying method and parameters, such as drying time, temperature, and relative humidity of the drying medium [16].

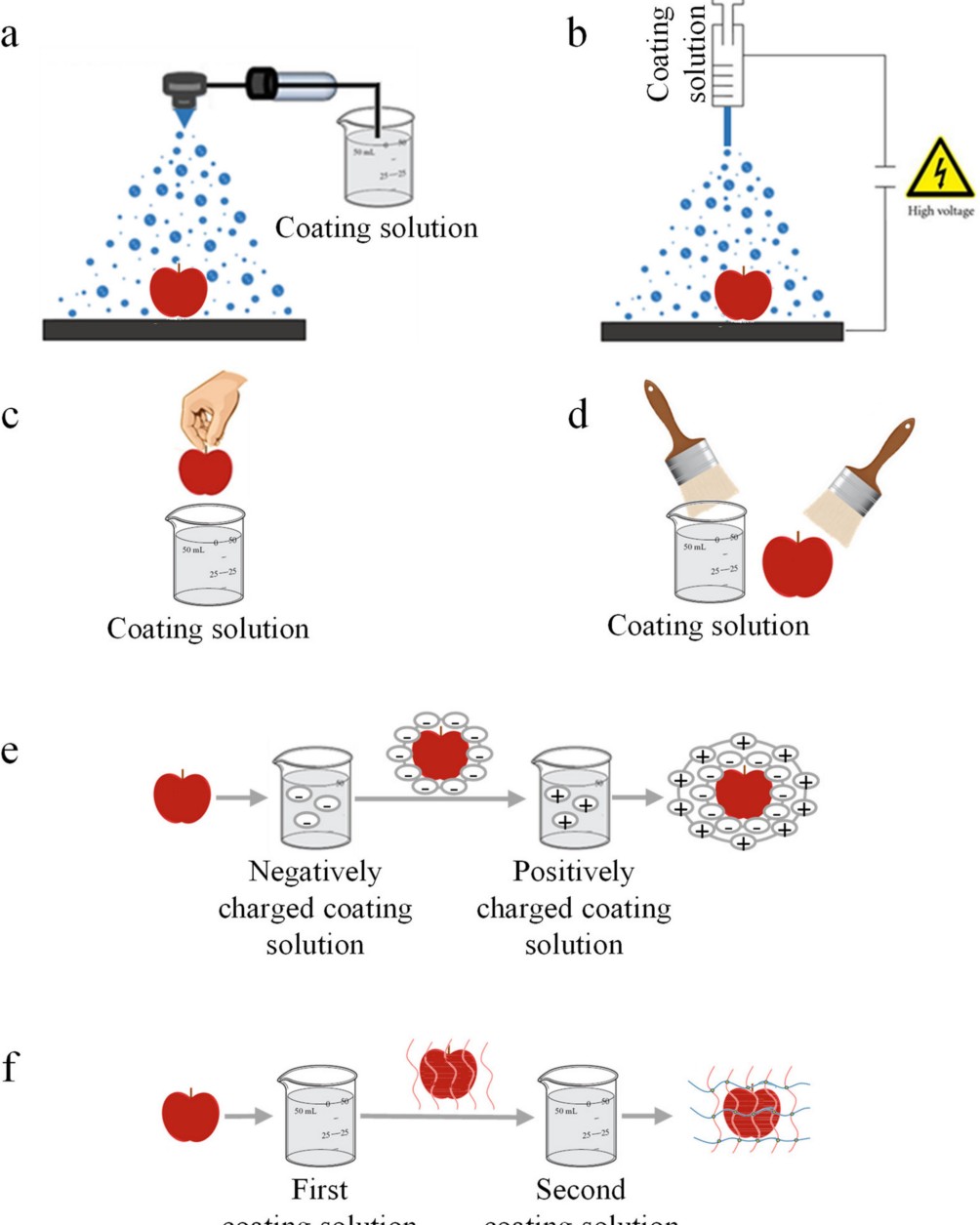

**Figure 1.** Schematic representation of the main coating deposition methods: (**a**) spraying method, (**b**) electrospraying method, (**c**) dipping method, (**d**) spreading method, (**e**) layer-by-layer coating deposition, and (**f**) cross-linked coating.

Among the advantages of spraying methods in coating application there are the absence of contamination of the coating solution, the possibility of continuous production automation and their suitability to be applied to heat-sensitive compounds [17].

To the best of our knowledge, no studies have been reported about the spraying deposition technique of NC-reinforced ECs on foods. Shanmugam and coworkers selected the spraying technique to deposit NC films by spraying on the surface of impermeable substrate, because they found that the spraying time was independent of NC suspension concentration [18]. However, to improve the smoothness of the obtained NC film, they preliminarily subjected the NC suspension to a high-pressure homogenization treatment to reduce fiber diameter and length. Other researchers reported that spraying a NC suspension on solid surfaces represent a valid alternative technique to vacuum filtration

to obtain high-quality thin films. Therefore, spraying offers significant advantages in specialized applications, such as contouring or contour coating (the coating solution freely falls across the substrate and forms a curtain, which follows the substrate contours) and contactless coating (no mechanical contact takes place between the base substrate and the distributor of the coating solution), especially because a high NC content can be usedthan in vacuu-filtration, hence reducing the amount of water to be removed by drying [19,20].

### 2.1.2. Electrospraying Method

A novel technique for coating application to food surfaces is electrospraying (Figure 1b), where the atomization of the coating solutions is carried out in a high-intensity electric field, which enables the formation of micrometric and sub-micrometric charged droplets with an extremely narrow size distribution [21]. The application of high voltage to the coating solution at the tip of an emitter causes the formation of a Taylor cone, with the accumulation of charge near the surface of the nascent droplet, and the destabilization of the liquid surface, which disrupts into multiple fine charged droplets [22,23]. Therefore, in comparison with spraying, which produces uncharged droplets, electrospraying offers the additional advantage of further promoting the adhesion to the food surface, due to electrostatic interactions [24]. During electrospraying, the droplet size, deposition rate, and layer thickness can be controlled by optimizing the main process parameters, such as coating solution flow rate and properties, namely conductivity and viscosity [25,26].

It must be remarked that, because of the low flow rate of the coating solution at each emitter (because of the sequential formation of individual, submicrometric droplets) and of the requirement for specialized personnel to work with the high-voltage generator, scaling-up of electrospraying is more expensive than spraying (multiple emitters are needed). In particular the electrospraying technique appears to be suitable especially for the fabrication of thin NC-based composite materials with oriented fibers, for tunable properties in an electric or magnetic fields (for example, using magnetic cellulose) [27].

### 2.1.3. Dipping Technique

The dipping method (Figure 1c) involves three important stages [28]:

1.  Immersion and holding (dwell time). The substrate is immersed into the coating solution, followed by a holding time to allow the substrate to interact for a sufficient dwell time with the coating solution to complete wetting.
2.  Deposition and drainage. By pulling the substrate upward, a thin layer of the coating solution is entrained, causing film deposition. In this stage, excess liquid drains from the surface of the substrate.
3.  Evaporation and/or drying. The excess diluent leaves the food surface by evaporation at room temperature or drying with heated air, thus achieving a thin film of the coating solution.

Previous studies have shown that the thickness and morphology of the coatings deposited by dipping on fruits, vegetables, meat, and fish significantly depends on immersion time, withdrawal speed, dip-coating cycles, density, viscosity and surface tension of the coating solution, substrate surface characteristics and drying conditions of the coating solutions [29]. The dipping method is especially advantageous when the coating has to be applied to products with a complex and rough surface, which cannot be uniformly reached by spraying methods. Dipping, however, suffers also from several drawbacks. First of all, dipping generally forms a thick coating layer, which may cause excessive reduction of product respiration and damage of food surface [30,31] and degradation of its functionality, reducing the storage characteristics [32]. Secondly, issues related to the contamination of the coating solution with bacterial load or dirtiness from processed fruit represents an important issue to consider in industrial scale-up. Finally, in general, large volumes of coating solutions are needed per unit mass of product to be coated, to ensure proper dipping conditions. Therefore, dipping is best suited to small-scale or batch processes.

The traditional way of ECs application is by dipping directly in a liquid form, forming a micro-layer film on the surface of the fresh food. However, from literature, only a few studies have used this technique for the deposition of NC-based coatings. For example, Herrera and coworkers verified the potential of NC-based coatings, applied by dipping, as an ecological bio-based option for developing barrier applications on paper-based packaging [33].

### 2.1.4. Spreading Method

The spreading method (Figure 1d), also known as brushing, is suitable for high-viscosity coating solutions that are spread directly onto the material surface and then dried. The main parameters used to characterize the spreading of the coating solution of the food surface are, generally, the wetting degree and the spreading rate [34]. The degree of spreading/wettability of a surface by a particular liquid is commonly evaluated by contact angle measurements, which are described in Section 4.1.3. The efficient coating deposition by spreading is affected by several factors, such as the substrate properties, and in particular surface roughness and geometry, liquid properties, such as viscosity, surface tension and density, and drying conditions, including temperature and relative humidity [35].

Brushing is generally carried out by specialized operators and, therefore, the quality of the spread coating and layer uniformity is strongly affected by the human factor. For all these considerations, spreading is more suitable for small-scale productions.

In literature, no applications have been reported for the spreading method for NC-based ECs deposition on food products. Nevertheless, the coating deposition by spreading seem to be a promising technique, when considering the numerous studies that have highlighted the effectiveness of NCs-reinforced films production through the conceptually similar film casting technique [36–42].

### 2.1.5. Layer-by-Layer Deposition

The application of ECs is often limited by the difficult adhesion of the coating solution to the product surface, especially in the case of fresh-cut fruits, characterized by high hydrophilic surfaces [16]. In the layer-by-layer (LbL) deposition method (Figure 1e), adhesion to the food surface is enhanced by electrostatic interaction of the food surface with charged polyelectrolytes. The electrostatic interactions are also exploited to form coatings made of two or more layers of nanometric dimensions, which are physically or chemically bonded to each other [43,44], enabling the efficient control of physicochemical properties and functionality of ECs.

The LbL electrostatic deposition technique generally relies on the combination of oppositely charged polyelectrolytes, through alternate dipping of the food product in different coating solutions. The alternate dipping process is repeated as many times as many coating layers are desired. The amount of adsorbed polyelectrolyte onto the food surface during each dipping depends on the ionic strength, pH, and charge densities of each coating solution.

LbL deposition methods do not only work with oppositely charged polyelectrolytes but also with macromolecules capable of developing hydrogen-bond, hydrophobic or covalent interactions through mutually interacting binding sites. Table 1 reports some examples of LbL coatings applied to fresh products exploiting the electrostatic interactions of the different biopolymer layers, with a special focus on the use of NC as anionic polyelectrolyte.

**Table 1.** Applications of electrostatic layer-by-layer (LbL) coating deposition methods for different food products.

| Product | Polyelectrolytes | | Results | References |
|---|---|---|---|---|
| | **Anionic** | **Cationic** | | |
| - | Nanocellulose | Nanochitin | Reinforcing film agent with excellent gas-barrier properties, highly transparent, unfavorable to bacterial adhesion and thermally recyclable, thus promising for advanced food packaging applications. | [45] |
| - | Nanocellulose | Chitosan, cationic starch and collagen | Ability to finely tailor the nanoarchitecture of the film providing ways high performance free-standing films or coatings with advanced properties. | [46] |
| - | Nanocellulose | Chitosan | Promising nanocomposite film with high oxygen barrier in transparent flexible packaging materials and semi rigid tridimensional objects. | [47] |
| - | Nanocellulose | Polyethyleneimine | Thin films with unique mechanical properties and the morphology of a "porous matchsticks pile", which brings about strong antireflective properties. | [48] |
| Fresh-cut apples | Carboxymethylcellulose sodium salt (NaCMC) | Chitosan | Polyelectrolyte multilayer (PEM) film shown good browning, weight loss, and metabolic activity inhibition ability. | [49] |
| Mandarin fruits | Carboxymethylcellulose (CMC) | Chitosan | The LbL polysaccharides-based coating notably improved the physiological quality of mandarins and their firmness. | [50] |
| Fresh-cut mangoes | Sodium alginate | Chitosan | Nanomultilayer coating by electrostatic self-assembly improved the microbiological and physicochemical quality of during storage time. | [51] |
| Citrus fruit | Carboxymethyl cellulose (CMC) | Chitosan | CMC/chitosan electrostatic bilayer EC greatly enhanced fruit glossiness and appearance but was not very effective in preventing weight loss. | [52] |
| Fresh-cut melons | Sodium alginate | Chitosan | LbL electrostatic deposition of ECs had benefits on food firmness, gas exchange, and microbiological protection | [53] |
| Mango fruits | Polystyrene sulfonate sodium salt (PSS) | Poly diallyl dimethylammonium chloride (PDADMAC) | PDADMAC/PSS films based-coated fruit shown significantly improved the hydrophilicity of the outer surface. | [54] |

LbL has drawn considerable attention because of its ability to control the thickness of the coating at the nanoscale, for the extensive choice of materials [55] and the possibility to embed antimicrobial components into the polymer matrix to construct the antimicrobial composites [56–58]. However, it must be remarked that, for industrial applications, its use is limited by the complex coating deposition procedure, based on alternating the use of

different coating solutions, often with the need for intermediate washing phases to remove excess coating solutions.

### 2.1.6. Cross-Linking Technique

The cross-linking technique can be described as the procedure of linking the polymer chains by covalent and non-covalent bonds. Cross-linked coatings (Figure 1f) are generally produced by deposition of the coating solution on the food surface by spraying, dipping, or spreading, followed by the deposition of a cross-linking agent, for the formation of a more compact and resistant coating. Cross-linked coatings offer significant advantages especially in the reduced migration of external molecules into the coatings [59,60], as well as in improving mechanical strength, chemical resistance, and the thermal stability of the coating [61]. The most common crosslinking agents are symmetrical bifunctional compounds with reactive groups with specificity for functional groups present on the matrix macromolecules [62]. Cross-linking is especially useful for biopolymer materials, such as those derived from proteins or polysaccharides, although it is more commonly applied to proteins than to polysaccharides since proteins have more functional groups [63]. The typical cross-linking process involves the use of a wide range of cross-linking agents (Table 2).

**Table 2.** Cross-linking agents commonly used in different types of edible coating.

| Cross-Linking Agent | Biopolymers | References |
|---|---|---|
| Glutaraldehyde | Gelatin<br>Cellulosic derivatives<br>Chitosan | [64]<br>[65]<br>[66,67] |
| Epichlorohydrin | Starch | [68,69] |
| $Ca^{2+}$ ions | Alginate<br>Pectin<br>Whey protein | [70–72]<br>[70]<br>[73] |
| Sodium benzoate | Starch | [74] |
| Citric acid | Starch<br>Cellulosic derivatives | [75,76]<br>[77,78] |
| Boric acid | Cellulose | [79] |
| Tannic acid | Chitosan<br>Gelatin | [80]<br>[81] |
| Ferulic acid | Gelatin | [81] |

The cross-linking technique can be applied also to the preparation of NC-based coatings, through the prior modification of NC. For example, nanocomposite films were prepared with corn nano-starch as the biopolymeric matrix and modified-CNCs as the reinforcement. The CNCs were modified through a two-step method, in which they were initially crosslinked with citric acid, and subsequently amidated with chitosan. The modified-CNCs loaded nanostarch-based nanocomposite film contributed to conferring (i) a stronger network structure through intra- and intermolecular hydrogen bonds and mutual entanglements with the starch matrix, (ii) an increase in tensile strength and water contact angle value, (iii) a decrease in water vapor permeability, (iv) a better antimicrobial activity against *E. coli* and *S. aureus* bacteria, when compared with the pure corn nano-starch film [39].

### 2.2. Optimization of Film-Forming Formulation

The efficiency of coating deposition is reported to depend primarily on the nature of the coating ingredients and their relative, optimal concentrations [82]. The optimization of the operating parameters for EC deposition can be supported by multivariate statistic tech-

niques, such as response surface methodology (RSM). RSM is a collection of mathematical and statistical tools based on the fit of a polynomial equation to the experimental data. It is intended to replicate the observed experimental behavior and help to derive statistical conclusions, to enable the reduction in the number of experimental runs normally required to assess the optimal values of multiple variables (multivariate analysis), especially in the case of significant variable interactions [83].

In the process of optimization of coating formulations, response variables are related to independent variables by a second-order polynomial equation (Equation (1)):

$$Y = \beta_0 \sum_{i=1}^{k} \beta_i X_i + \sum_{i=1 \ i<j}^{k-1} \sum_{i=2}^{k} \beta_{ij} X_i X_j + \sum_{i=1}^{k} \beta_{ii} X_i^2 \tag{1}$$

where $X_i$ is independent variables, $\beta_0$ the intercept; $\beta_i$, $\beta_{ii}$, $\beta_{ij}$ are regression coefficients linear, quadratic, and interaction terms, respectively, and k is the number of variables.

Different response variables were used in coating optimization. For example, RSM was applied to optimize gelatin/chitosan solutions' concentration in film in terms of finding the maximum elasticity by minimizing Young's modulus [84]. It is also used, for example, to optimize the sodium alginate and calcium chloride concentrations and dipping time to minimize the coating thickness [85]. In another study, the process variables for the preparation of edible composite films from pearl millet starch and carrageenan gum blends were optimized as a function of coating quality, evaluated in terms of thickness, water vapor permeability, solubility, and tensile strength [86]. The response surface methodology was used also to estimate the effects of the independent variables, such as alginate, glycerol, and citric acid concentrations on the surface solid density of coated papaya [87]. Color, water content, water-solubility, puncture strength, percentage of elongation, and water vapor permeability of coatings were also evaluated. RSM was also used to determine the relationships that both turmeric oil volume and coating thickness have with the antimicrobial agent's migration rate, the microbial inhibition zone and the degree of weight loss during biodegradation [88].

The use of NC as a reinforcement to improve the performance of ECs, as extensively described in Section 4, requires the optimization of NC concentration, to avoids its aggregation in the film-forming solutions. The optimization of NC, polymer and plasticizer concentrations through RSM has been previously carried out by investigating their combined effect on the mechanical properties of the resulting coating (Young modulus, tensile strength at break and strain analysis at break) [89].

## 3. Classification and Properties of Nanocellulose

Cellulose, which is the most abundant carbohydrate polymer on earth, is characterized by noteworthy structure and properties. This renewable natural biopolymer, together with the materials deriving from it, has attracted considerable interest, especially for application in environmentally friendly and biocompatible products and in foods [90,91]. Cellulose exhibits a unique molecular structure, consisting of a linear homopolysaccharide composed of glucose monomers, linked together by β-1-4-glycosidic bonds, which confers unique properties, such as hydrophilicity, chirality, degradability, and broad chemical variability initiated by the high donor reactivity of the OH groups. Moreover, cellulose isolation and modification, especially through advanced nanotechnology tools, enabled further promotion of its techno-functional attributes [92]. Owing to their hierarchical order in a supramolecular structure and organization given by the hydrogen bond network between hydroxyl groups, nanoparticles can be efficiently isolated from cellulose [93] via mechanical and chemical methods, or through their combination. The various types of cellulose nanoparticle (also known as nanocellulose, NC) can be classified based on their shape, dimension, function, and preparation method, which in turn primarily depend on the cellulose origin, the isolation and processing conditions as well as the eventual pre- or post-treatment [94,95]. The physicochemical characteristics of cellulose at the nanoscale, such as

high specific surface area and aspect ratio, high crystallinity, purity, excellent mechanical properties, and low thermal expansion and density [95–101], open new prospects for NC use in several fields, including biomedical, environmental, and energy applications [102]. The cellulosic materials having at least one dimension in the nanometer range, based on structure and particle diameters [103], is usually classified into cellulose nanocrystals (CNC), cellulose nanofibers (CNF) and bacterial nanocellulose (BNC). CNC and CNF can be extracted through a top-down process, whereas BNC is synthesized through a bottom-up approach [104].

### 3.1. Cellulose Nanocrystals (CNC)

CNC are renewable bio-based nanoparticles with a rod-like shape and at least one dimension below 100 nm [105]. CNC are usually isolated from crystalline cellulose microfibrils upon treatment with acid at high temperature [106]. CNC have many attractive characteristics, such as high mechanical strength, high aspect ratio (having mean diameters of 2–20 nm and lengths of 100–500 nm), lightweight, biodegradability, good biocompatibility, and potential for surface chemical modifications [107–109]. These distinctive features make CNC a promising material for numerous applications, especially in packaging materials, biomedical engineering, food emulsions, biosensors, hydrogel systems, and water purification [110]. Table 3 reports the most recent advances in the application of CNC isolated from different raw materials and with different morphological and shape characteristics.

**Table 3.** Recent advances in the production process, physicochemical characteristics and application of CNC extracted from different sources (literature data for years 2020 and 2021).

| Source | Production Process | Morphology/Shape | Dimensions | Crystallinity | Applications | References |
|---|---|---|---|---|---|---|
| Pine | | Spherical morphology | 50–100 nm diameter | 55% | | |
| Teak | Acid hydrolysis | Rod-like surface topographies | 50–60 nm diameter | 52% | – | [111] |
| Sugarcane bagasse | | Rod-like structure | 20–60 nm in diameter | 45% | | |
| Eucalyptus pulp | Acid hydrolysis | Rod-like structure | 130–250 nm in length and 15–30 nm in diameter | – | Starch based composite film | [112] |
| Waste cotton fibers | Ultrasound-assisted acid hydrolysis | Short rod shape | 200–500 nm length and 10–15 nm diameter | 86% | PLLA/PDLA composites films | [113] |
| Commercial microcrystalline cellulose | Alkali hydrolysis followed by ultrasound-assisted acid hydrolysis | Spherical shape | 30–60 nm in diameter | 81% | Stabilizer for Pickering emulsions | [114] |
| Water hyacinth stem fiber | Acid hydrolysis | Spherical-like particles | 20–50 nm in diameter | 72% | Reinforcement for polyvinyl alcohol (PVA)-gelatin nanocomposite | [115] |
| Commercial microcrystalline cellulose | Acid hydrolysis | Spherical shape | 126–134 nm length and 3–11 nm diameter | 77%–83% | Pickering emulsion stabilizers and surface cleaning agents | [116] |

**Table 3.** *Cont.*

| Source | Production Process | Morphology/Shape | Dimensions | Crystallinity | Applications | References |
|---|---|---|---|---|---|---|
| Enteromorpha Ulva prolifera green seaweed | Acid hydrolysis | – | – | – | Reinforcement for chitosan-ulvan hydrogel | [117] |
| Cellulose-rich cotton fibers | Alkali hydrolysis followed by acid hydrolysis | Bundles of rod-like particles | 60 nm in lenght | 89% | Reinforcement for chitosan-ulvan hydrogel | [118] |
| Cotton | Ultrasound-assisted acid hydrolysis | Spherical rod-like shape | 50 nm in diameter | 81% | – | [119] |
| Commercial cellulose | Acid hydrolysis | Ribbon-like structure | 173 $\pm$ 6.3 nm in length and 10 $\pm$ 0.4 nm in diameter | 81% | Reinforcement for waterborne polyurethanes | [120] |
| Commercial cellulose | Acid hydrolysis | Rod-like particles | 128 $\pm$ 55 nm in length and 14 $\pm$ 4 nm in diameter | 84% | Tunable nanomaterial for pervaporation membranes based on a hydrophobic poly(styrene)-poly(butadiene)-poly(styrene) (SBS) matrix | [121] |
| Paper powders | Acid hydrolysis | Rod-like particles | 100 nm in length and 7 nm in diameter | 65% | Reinforcement for polyurethane (PU) nanocomposites for medical applications | [122] |
| Sawdust | Ultrasound pre-treatment followed by aid hydrolysis | Dot-like shape | 6 nm in diameter | – | Polyamide thin-film composite membranes for enhanced water recovery | [123] |
| Jute fibers | Acid hydrolysis followed by alkali hydrolysis | Rod-like structure | 400–1200 nm length and 40–90 nm diameter | – | Reinforcement for pSiDm hydrogel to treat waste effluent | [124] |
| Palm fibre | Acid hydrolysis | Rod-like shapes | – | 84% | Potential filling agent | [125] |

### 3.2. Cellulose Nanofibers (CNF)

CNF are characterized by very different structures and properties than CNC, thus, defining different application areas [94]. CNF are composed of stretched bundles (aggregates) of elementary nanofibrils constructed from alternating crystalline and amorphous domains. Unlike CNC, the nanofibrils can contain a considerable non-crystalline fraction, with their crystallinity typically in the range of 50%–65%. CNF have lateral size of several tens of nanometers and length of few microns and, therefore, the aspect ratio of CNF is relatively large [7]. CNF have been isolated through different mechanical disintegration methods, such as high-pressure homogenization, ultrasonication, microfluidization, grinding, cryo-crushing, ball milling, and extrusion [101] or mechanical treatment in combination with chemical or enzymatic hydrolysis. Owing to their high aspect ratio and entanglement, cellulose nanofibers have the potential to be used in many different areas (see Table 4), particularly as strong reinforcement in development of nanocomposites [126,127].

**Table 4.** Recent advances in the production process, physicochemical characteristics and application of CNF extracted from different sources (literature data for years 2020 and 2021).

| Source | Production Process | Morphology/Shape | Dimensions | Crystallinity | Applications | References |
|---|---|---|---|---|---|---|
| Waste cotton fibers | Ultrasound-assisted acid hydrolysis | Fibrous | 15–20 nm in width and 1000–3000 nm in length | 79% | PLLA/PDLA composites films | [113] |
| Sugarcane bagasse | $(NH_4)_2HPO_4$ phosphorylation and mechanical high-speed blending | Fiber bundles | $18 \pm 9$ μm in width and $458 \pm 130$ μm in length | 69% | Gel | [128] |
| Bleached pulp paper | Enzymatic pre-treatment and then a high-pressure homogenization step | Fiber bundles | 28.1 nm in diameter and 4.9 μm length | – | Stabilization of the emulsion of Alkenyl Succinic Anhydride in water | [129] |
| Birch fibers | Microfluidizer assisted TEMPO-mediated oxidation | – | – | – | Reinforcement for hydrogels | [130] |
| Recycled milk-container board | Deep eutectic solvent treated and mechanical grinding | – | 2–80 nm in diameter | – | Filter material for aerosol filtration | [131] |
| Rice straw | Alkaline hydrolysis, bleaching and TEMPO-mediated oxidation | Homogeneous fibril structure | 5–10 μm diameter and 10–40 nm width | – | Composite membrane to increase electrochemical performance of supercapacitor | [132] |
| Wood pulp sheets Bamboo pulp sheets Low lignin-containing bamboo pulp sheets Bamboo powder | $(NH_4)_2HPO_4$ phosphorylation and mechanical ultra-fine grinder | Soft fiber structure Rod-like structure | 10–20 μm in diameter | – | Cellulose-based film for flame-retardant packaging materials | [133] |
| Commercial microcrystalline cellulose | Ultrasonic treatment following sulfuric acid hydrolysis | Beads-on-a-string cellulose nanofibril | 10–30 μm width and 40–50 μm length | 77% | Gelatin composite hydrogels | [127] |
| Licorice residues | Alkali and enzymatic hydrolysis followed by high-pressure homogenization | Nanofiber structure | 130 nm in diameter and 8 μm in lenght | – | Nanocomposite film | [134] |
| Commercial chitosan powder | High-pressure homogenization assisted TEMPO-mediated oxidation | | 204 nm in diameter and 13 μm in lenght | | | |
| Maize stalk waste residues | Mechanical grinding assisted chemical treatments | Highly entangled fibres network and web like structure | $35.48 \pm 12.60$ nm in diameter | 71% | Reinforcement material for biopolymer films for food packaging applications | [135] |

### 3.3. Bacterial Nanocellulose (BNC)

BNC has the same chemical structure as plant cellulose, i.e., is a linear hompolymer of repeating subunits β(1,4)-D-glucose with the molecular formula $(C_6H_{10}O_5)_n$. Compared to plant cellulose, BNC is chemically pure since it is free from hemicellulose, pectin and lignin. The synthesis of BNC occurs via cellulose synthase enzyme at cytoplasmic membrane level by several microbial genera belonging to *Acetobacter*, *Achromobacter*, *Bacillus*, *Sarcina*, *Aerobacter*, *Agrobacterium*, *Escherichia*, *Azotobacter*, *Rhizobium*, *Enterobacter*, *Klebsiella*, *Salmonella* [136–138]. Due to the standardized high molecular structure and inherent nanostructure, BNC possesses multifunctionality and good mechanical properties [139]. It is generally characterized by good hydrophilicity, high water-holding capacity, slow water release rate, high degree of crystallinity, and ultrafine fiber network [102,140,141]. The properties of BNC depend not only on its species of origin but also on the used substrate,

cultivation mode and cultural parameters. In addition to its multiple unique features, BNC also belongs to the category of generally recognized as safe (GRAS) products, and, therefore, it is widely used in food industry, biomedical, and pharmaceutical, as summarized in Table 5.

**Table 5.** Recent advances in the production process, physicochemical characteristics and application of CNF extracted from different bacterial sources and substrates (literature data for years 2020 and 2021).

| Source | Production Process | Morphology/Shape | Dimensions | Crystallinity | Applications | References |
|---|---|---|---|---|---|---|
| Bacterial cellulose pellicles | Acid hydrolysis and ultrasonic treatment | Rod or needle-shaped nanocrystals | 15–56 nm in width and 259–1142 nm in length | 83% | Nisin-loaded BCNs as antimicrobial agents in active food packaging | [140] |
| Pellicle-shaped bacterial cellulose | Mechanically defibrillation and acid hydrolysis | Rod-type crystal morphology | 20–30 nm in diameter | - | Reinforcement for sericin film | [141] |
| Bacterial cellulose | 2,2,6,6-tetramethylpiperidine-nitrogen-oxide (TEMPO) oxidation | Fibrils bundles | 70–100 nm in width | - | O/W Pickering emulsion stabilizer | [142] |
| Bacterial cellulose pellicles from organic waste and kombucha | Fermentation using glycerol as carbon source | 3D structure of cellulose fibrils | 100–2000 nm in length and 5 nm in width | 64%–80% | Composites | [143] |
| Bacterial cellulose | 2,2,6,6-tetramethylpiperidine-1-oxyl radical (TEMPO) oxidation | Nanofibrils | 5–10 nm in width | - | Pickering emulsion system stabilizer | [144] |
| Bacterial cellulose pellicles from grape pomace | Fermentation using carbon and nitrogen source | Ribbon-shaped cellulose nanofibers and nanofiber aggregates | 18–57 nm in width and micrometers in length | 68%–85% | Nanoadditives for oil well cement cement | [145] |
| Bacterial cellulose | High-pressure homogenization treatment | Nanofibrils | 97 nm in width and 6 nm in height | - | Pickering emulsion stabilizer | [146] |
| SCOBY, black tea | Fermentation | Nanofibers | 20–100 nm in diameter | 73%–79% | Reinforcement for chitosan nano-biocomposite films | [147] |
| Bacterial cellulose | Alkaline treatment | Tangled fibers | 50.73–140.25 nm in diameter | 84%–88% | Small-caliber vascular grafts | [148] |
| Bacterial cellulose | Fermentation in static culture | Ribbon-shaped fibrils | 70–80 nm in width | - | Reinforcement for film with carbon dots | [149] |

## 4. Characterization of Nanocellulose (NC)-Reinforced Coatings

This section provides a brief review of the analytical measurements that are routinely used to assess the successful coating deposition, as well as its reproducibility, for meeting the required specification for industrial applications.

### 4.1. Physical-Chemical Properties

#### 4.1.1. Thickness Determination

The coatings thickness represents an important factor when selecting or optimizing a deposition process for a particular application [150]. In addition to determining the acceptability of the coating process, it affects also the coating functionality, particularly permeability to water and gases [151].

The coating thickness is a function of the coating solution properties, such as polymer concentration (see Table 6), density, viscosity, and surface tension, as well as of the operating

parameters of deposition, such as, for example, the surface withdrawal speed for dipping deposition [152]. It can be determined by peeling the coating from the surface of the coated product and proceeding to the direct measurement of the film thickness using a micrometer screw gauge, simply known also as a micrometer. When peeling is difficult, for example in the case of the very thin coating layers obtained by the LbL method, in situ techniques can be applied, such as confocal Raman microspectrometry (CRM), surface-enhanced Raman scattering (SERS), and Fourier transform (FT)-Raman spectrometry [153,154].

**Table 6.** Typical values of tensile strength and percent elongation at break of commonly used edible coatings and films, as a function of the film-forming material, its concentration, and resulting thickness.

| Film-Forming Material | Concentration (% *w/w*) | Thickness (µm) | Mechanical Properties | | References |
|---|---|---|---|---|---|
| | | | Tensile Strength (MPa) | Elongation at Break (%) | |
| Agar | 1–3 | 31.2–70.2 | 14.3–37.4 | 12.4–31.8 | [155] |
| Starch | 5 | 200 | 1.41–8.03 | 12.97–56.25 | [156] |
| Alginate | 1.5 | 26.2–38.9 | 44–52 | 12.1–16.4 | [157] |
| Cellulose | 5 | 500 | 25 | 7 | [158] |
| Chitosan | 1.5 | 14.4–16.2 | 47.8–58.2 | 27.7–36.1 | [159] |
| Carrageenan | 2.5 | 51.6–64.8 | 40 | 20 | [160] |
| Gums | 10 | - | 3.5 | 60–80 | [161] |
| Pectin | 3 | 36 | 42–82 | 12–28 | [162] |
| Proteins | - | - | 3.3–3.9 | 160-213 | [163] |

In general, the addition of nanocellulose into nanocomposite coatings results in a slight increase in thickness, mainly related to the higher solid content in the coating solutions and the interruption of the original polymeric structure by NC, as extensively shown in Table 7. Therefore, the effect of NC incorporation on coating thickness can be correlated well with the concentration of NC in the formulation [36–38,42,164].

**Table 7.** Effect of nanocellulose (NC) on thickness and mechanical properties of edible coatings and films.

| Film-Forming Material | Cellulose | | Thickness (µm) | Mechanical Properties | | References |
|---|---|---|---|---|---|---|
| | Type | Concentration (% *w/w*) | | Tensile Strength (MPa) | Elongation at Break (%) | |
| Chitosan | CNF | 1.5 | 14.5–21.2 | - | - | [165] |
| Tapioca, potato, corn | CNF | 0 | 2.99 | 0.047 | 6.67 | [166] |
| | | 10 | 6.33 | 0.055 | 22.67 | |
| | | 20 | 5.71 | 0.056 | 30.51 | |
| Faba bean protein isolate | CNC | 0 | - | 4.3 | 105.0 | [167] |
| | | 1 | | 4.2 | 61.3 | |
| | | 3 | | 3.8 | 48.1 | |
| | | 5 | | 5.3 | 48.2 | |
| | | 7 | | 6.5 | 46.3 | |
| Cassava starch | Microcrystalline cellulose | 0 | - | 7.15 ± 0.6 | 22.75 ± 2.34 | [168] |
| | | 0.14 | | 8.19 ± 0.9 | 19.23 ± 2.25 | |
| | | 0.3 | | 9.91 ± 0.7 | 5.85 ± 1.43 | |
| | | 0.6 | | 10.99 ± 0.5 | 1.31 ± 0.25 | |
| Okara soluble dietary fiber and pectin | Sodium carboxymethyl cellulose | 0.5 | 123 ± 70 | 6.567 ± 0.33 | 16.67 ± 0.35 | [169] |
| Konjac glucomannan | BNC | 0 | 39 ± 6 | 46.43 | 6.34 | [170] |
| | | 1 | 40 ± 12 | 50.36 | 8.58 | |
| | | 2 | 41 ± 0 | 69.29 | 9.44 | |
| | | 3 | 41 ± 15 | 74.05 | 8.18 | |
| | | 4 | 42 ± 10 | 82.01 | 5.70 | |
| Cassia-gum | Carboxylated CNC | 0 | 89 ± 5 | 18.53 | 28.87 | [36] |
| | | 2 | 90 ± 3 | 24.77 | 31.88 | |
| | | 4 | 93 ± 2 | 32.85 | 34.75 | |
| | | 6 | 98 ± 4 | 28.75 | 36.51 | |
| Polyvinyl alcohol | NC | 1 | - | 6.42 ± 0.59 | 89.99 ± 11.77 | [171] |
| | | 3 | | 9.47 ± 1.62 | 106.94 ± 7.04 | |
| | | 5 | | 11.17 ± 1.08 | 117.52 ± 10.28 | |

**Table 7.** *Cont.*

| Film-Forming Material | Cellulose | | Thickness (μm) | Mechanical Properties | | References |
|---|---|---|---|---|---|---|
| | Type | Concentration (% *w/w*) | | Tensile Strength (MPa) | Elongation at Break (%) | |
| κ-carrageenan | CNC | 0 | 20 | 38.33 ± 3.79 | 21.50 ± 3.72 | [172] |
| | | 1 | 30 | 38.43 ± 5.94 | 22.93 ± 1.50 | |
| | | 3 | 40 | 39.83 ± 0.38 | 23.83 ± 2.71 | |
| | | 5 | 25 | 40.07 ± 2.80 | 24.33 ± 3.00 | |
| | | 7 | 25 | 52.73 ± 0.70 | 28.27 ± 2.39 | |
| | | 9 | 35 | 39.10 ± 1.04 | 25.83 ± 2.61 | |
| k-CA biopolymer | CNC | 0 | | 49.0 | 27.5 | [40] |
| | | 1 | | 59.2 | 23.1 | |
| | | 3 | 80 | 66.6 | 20.7 | |
| | | 5 | | 80.9 | 18.9 | |
| | | 8 | | 85.1 | 15.4 | |
| Whey protein | CNC | 0 | | 1.30 | 47 | [41] |
| | | 1 | | 1.65 | 35 | |
| | | 2 | | 2.04 | 33 | |
| | | 3 | | 2.10 | 34 | |
| | | 4 | - | 2.29 | 35 | |
| | | 5 | | 2.30 | 35 | |
| | | 10 | | 2.70 | 25 | |
| | | 15 | | 3.15 | 24 | |
| Corn nanostarch | CNC | 0 | | 3.41 ± 0.17 | | [39] |
| | | 0.2 | | 5.99 ± 0.30 | | |
| | | 0.4 | | 7.28 ± 0.36 | | |
| | | 0.6 | 300 | 8.61 ± 0.43 | - | |
| | | 0.8 | | 11.25 ± 0.56 | | |
| | | 1 | | 7.78 ± 0.39 | | |
| Agar | BNC | 0 | | 22.10 ± 0.64 | 10.76 ± 2.30 | [173] |
| | | 0.045 | | 27.95 ± 1.42 | 14.50 ± 0.88 | |
| | | 0.075 | - | 31.26 ± 2.26 | 27.47 ± 1.08 | |
| | | 0.12 | | 34.20 ± 1.35 | 21.53 ± 1.62 | |
| | | 0.15 | | 44.51 ± 1.86 | 13.02 ± 1.70 | |
| Whey protein | CNC | 0 | | 2.30 ± 0.35 | 46.07 ± 23.25 | [174] |
| | | 2 | - | 3.41 ± 0.87 | 20.82 ± 9.85 | |
| | | 5 | | 3.49 ± 0.91 | 26.54 ± 9.12 | |
| | | 8 | | 4.93 ± 0.49 | 17.63 ± 3.93 | |
| Chitosan | BNC | 0 | 90 | 21.07 ± 1.64 | 33.84 ± 2.51 | [37] |
| | | 2 | 100 | 27.03 ± 1.46 | 29.71 ± 2.15 | |
| | | 4 | 100 | 41.32 ± 2.20 | 23.76 ± 1.52 | |
| | | 6 | 110 | 34.75 ± 1.02 | 25.11 ± 2.93 | |

### 4.1.2. Mechanical Properties

ECs must resist breakage and abrasion during food handling. Moreover, they must also exhibit adequate flexibility to adapt to possible food deformation during storage without breaking, while still protecting the food. The mechanical properties of films and coatings are generally characterized through two main parameters, tensile strength, and percent elongation at break, determined as specified in the standards of the American Society for Testing and Materials (ASTM). The mechanical properties of ECs depend on the type of film-forming material, its concentration, and production technique [175] (Table 6).

Moreover, different studies demonstrated that the incorporation of NC in the film-forming material enhanced the coating mechanical properties (Table 7), by altering the internal structure and intensifying the interaction forces [176,177].

### 4.1.3. Surface Wettability

The effectiveness of ECs for food protection depends on the uniformity of wetting and spreading on the surface of the fresh produce and, after drying, depends on their adhesion, cohesion, and durability [178]. The effective spreading of a coating solution on the surface of food depends on the wettability of the coating solutions on the food surface and can be correlated with the resulting coating thickness and consequent biological properties and shelf life of the coated product.

When a drop of liquid is placed on a solid surface, the liquid is subjected to the balance between adhesive and cohesive forces, where adhesive forces cause the liquid to spread over the solid surface, while cohesive forces cause it to shrink [32]. The wetting of the liquid

on the solid surface can be evaluated through contact angle measurements, which assess the mechanical equilibrium of the drop under the action of three interfacial tension forces, at solid–vapor, solid–liquid, and liquid–vapor interfaces, according to the equilibrium relation known as Young's equation [179]. The ideal case of a contact angle value equal to 0° corresponds to a hydrophilic solid surface where total wetting conditions can be attained by an aqueous solution. A contact angle value comprising between 0° and 180° suggests the occurrence of partial wetting, which is higher for the contact angle below 90°. The ideal case of a contact angle equal to 180° corresponds to a hydrophobic solid surface, where no wetting conditions occur when in contact with an aqueous medium. The contact angle can be measured directly on the food surface through the sessile drop method [180,181] or atomic force microscopy (AFM) [182].

### 4.1.4. Barrier Properties

The efficiency of ECs strongly depends also on their barrier properties to the permeation of gas, water vapor, aroma, and oil. The barrier properties can be considered as a function of the chemical composition and structure of the coating-forming polymers, the characteristics of the product, and the storage conditions [16]. The mass transport properties through the coating can be described by three principal mechanisms [183]:

1. Diffusion. It is the rate of movement of a permeant molecule through the tangled polymer matrix, based, for example, on the size of the permeant molecule and the structure of the polymer matrix. Molecular diffusion through a film generally obeys Fick's first law in one dimension, as described by Equation (2):

$$J = -D\frac{\partial C}{\partial x} = D\frac{C_1 - C_2}{l} \tag{2}$$

   where $J$ is the molecular diffusion of the permeant molecule, $D$ is its diffusion coefficient and $C$ its concentration, $l$ is the thickness of the edible film, and subscripts 1 and 2 refer to the internal and external sides of the coating.

2. Solubility. This is the partitioning behavior of a permeant molecule between the surface of the polymer and the surrounding headspace. The solubility coefficient $C$ can be defined by Henry's law, as shown in Equation (3):

$$C = S \times P \tag{3}$$

   where $S$ is the solubility coefficient of the permeant molecule, and $P$ is the environmental pressure.

3. Permeability. This is the rate of transport of a permeant molecule through the polymeric layer as a result of the combined effects of diffusion ($D$) and solubility ($S$). Therefore, the permeability coefficient ($\Pi$), which characterizes the intrinsic permeability of the edible film, can be described as shown in Equation (4):

$$\Pi = D \times S \tag{4}$$

With the assumptions that the diffusion occurs in a steady-state, and the diffusivity coefficient is constant, the molecular flux ($J$) can be expressed through Equation (5):

$$J = D\frac{C_1 - C_2}{l} = \frac{\Delta m}{A \times \Delta t} \tag{5}$$

where $\Delta m$ is the amount of vapor or gases diffusing through a film of area ($A$), during a finite time ($\Delta t$). The application of Henry's law (Equation (3)) allows expression of the driving force in terms of partial pressure (($C_1 - C_2$) = $S \times \Delta P$). Rearrangement of terms,

and expressing the diffusivity as a function of the permeability coefficient (Equation (4)), yields the following Equation (6):

$$J = \Pi \frac{\Delta P}{l} = \frac{\Delta m}{A \times \Delta t} \tag{6}$$

Then, the permeabilities of $O_2$, $CO_2$, and water vapor can be calculated by the following equation [184].

$$\Pi = \frac{\Delta m}{\Delta t} \frac{l}{A \times \Delta P} \tag{7}$$

Factors affecting a polymer's structure have a direct effect on segmental mobility and, therefore, influence its mass transport properties [183]. Several polymer properties influence permeability: chemical structure, method of polymer preparation, polymer processing conditions, free volume, crystallinity, polarity, tacticity, cross-linking and grafting, orientation, presence of additives, and use of polymer blends [185].

The incorporation of NC in coating solutions is generally reported to significantly affect the barrier properties of the films. In some polymeric matrices, the transmission rate of water vapor was reported to increase with NC addition, because of (i) the increase of hydrophilicity within the polymer [38]; (ii) the change in polymer adsorption since the crystallinity, internal structure and interaction forces are changed [42]; (iii) the higher concentration of NC which causes its agglomeration in the film matrix [36]. However, in other cases, the water–vapor barrier properties were reported to increase, because of the increased surface–volume ratio and compactness of film network [37], due to the formation of a network of hydrogen bridges between NC and the polymeric matrix, which resulted in a winding path for the water molecules, hindering their propagation through films [36,39]. The effect of NC incorporation in film-forming solution on the barrier properties can, therefore, be related to the chemical nature of NC (chemical structure, polarity, degree of crystallinity) and its concentration, as well as the hydrophilicity and hydrophobicity of the film matrix.

### 4.1.5. Optical Properties

The appearance of the coated food products affects consumers' acceptance. Therefore, coating optical properties, such as color, gloss, and transparency, also need to be optimized. The parameters that mainly affect the optical properties of the coating layer can be reported in terms of its internal and surface microstructure. The intensity of light reflected by the coated food can be determined in terms of the light directly reflected at the interface between air and the coated food surface (specular reflection), and by the light re-emitted out of the surface in all the directions after penetrating the coating of the food and scattering internally (indirect reflection) [16].

The transparency of ECs depends on their internal structure, which is affected by the film-forming compositions and concentrations, particle size distribution and rearrangement during drying, due to destabilization phenomena such as creaming, aggregation and/or coalescence [186]. The incorporation of NC as reinforcement material commonly decreases the transparency of coatings, hence, causing higher opacity than control films [41]. This is due to the strong interaction between NC and the polymeric matrix, as well as to the light dispersion effect from added NC [187]. The transparency of films can be measured through the Kubelka–Munk theoretical model [188]. This theory models the reflected and transmitted spectrum of a colored layer based on a material-dependent scattering and absorption function, with the following assumptions [189]:

1.  A translucent colorant layer on the top of an opaque background;
2.  Within the colorant layer, both absorption and scattering occur;
3.  The light within the colorant layer is completely diffuse.

The gloss of films is affected by their microstructure and depends in particular on the type and concentration of surfactant, diameter and particle size distribution of the

dispersed phase, relative humidity, storage time and surface roughness [186,190–192]. Nevertheless, other factors such as the angle of incident light or the intrinsic properties (refractive index) of the material also affect the film gloss [193].

The color can be evaluated through a colorimeter or a spectrophotometer. The parameters denoting luminosity ($L^*$), red-green hue ($a^*$) and yellow-blue hue ($b^*$) are the edible film color values in the CIElab color space. The main color parameters used in evaluating the optical properties of the films [194] are reported in the following:

- Color difference $\Delta E$ (Equation (8)):

$$\Delta E = \sqrt{\Delta L^2 + \Delta a^2 + \Delta b^2} \tag{8}$$

- Chrome $C$ (Equation (9)):

$$C = \sqrt{(a^*)^2 + (b^*)^2} \tag{9}$$

- Hue angle $H$ (Equation (10)):

$$H = arctan\left(\frac{b^*}{a^*}\right) \tag{10}$$

- Whiteness index $WI$ (Equation (11)):

$$WI = 100 - \sqrt{(100 - L^*)^2 + (a^*)^2 + (b^*)^2} \tag{11}$$

where $\Delta L$, $\Delta a$, and $\Delta b$ are the differences of $L^*$, $a^*$, and $b^*$ with the standard color of a white disk ($L^*_0$, $a^*_0$, and $b^*_0$, respectively).

### 4.1.6. Microstructure

Properties of ECs depend on several factors, such as the ratio of crystalline to amorphous zones, polymeric chain mobility, and specific interactions between functional groups of polymers and the permeant substance within amorphous zones. Common techniques used to elucidate the coating microstructure include scanning electron microscopy (SEM), Fourier transform infrared spectroscopy (FTIR), X-ray diffraction, differential scanning calorimetry (DSC), thermo-mechanical analysis (TMA), and dynamic mechanical analysis (DMA) [195]. SEM may be useful to evaluate film homogeneity, layer structure, the morphology of pores and cracks, surface smoothness and thickness [196]. FTIR may be used to evaluate the extent of interactions between the different film components [197]. X-ray diffraction may provide an estimate of the amorphous-crystalline structure of film polymers and to track recrystallization during storage [198]. The evolution of the crystalline structure in the coating matrix during storage can be evaluated by DSC. Both DSC and TMA techniques are commonly used to estimate the glass transition temperature, which is strongly dependent on both the film composition and moisture content and, therefore, can be correlated with the stability of a polymeric film [199].

Generally, films containing NC as a reinforcement additive presented characteristics of being homogenous, continuous, having a smooth surface without pores or granules and bubble-free, indicating good NC dispersion in the polymeric matrix. However, when the concentration of NC was increased, the roughness of the cross-section of the films also increased [172,187,200–203]. This can be ascribed to the aggregation of nanoparticles due to their high hydroxyl content [200,204], to the hydrogen bonds and electrostatic interactions between NC and polymers, tightening the network, resulting in smaller pores, and making the materials less homogeneous and more opaque [202]. The microstructure change is reflected by the reduction of barrier properties, since the formation of paths may facilitate the passage of water vapor, as previously reported. The compatibility of the materials was attributed to factors such as (i) chemical similarities between starch and

cellulose, (ii) interaction of hydrogen bonds between NC and the matrix and (iii) effect of NC nanometric size [187].

### 4.2. Antimicrobial Properties and Shelf-Life Extension

ECs are usually applied on highly perishable products, such as fresh and fresh-cut fruits and vegetables, to extend the shelf life and to preserve their quality and minimize losses through controlling physiological, biochemical or oxidation processes. To enhance their efficiency and functionality, ECs can be loaded with different bioactive compounds (as illustrated in Figure 2) to develop specific functionalities, such as antimicrobial, anti-browning, antioxidant, coloring, and flavoring, or even nutritive actions [205].

The addition of antimicrobial agents to the coating solution is reported to develop a synergistic action with the physical barrier of the coating. Moreover, the incorporation in the coating layer might also enable the controlled release of the antimicrobial molecules on the food surface [16,206–208], contributing to improving the shelf-life of the product, by inhibiting the growth of bacterial and fungal cells over an extended time. In contrast, the direct use of antimicrobials in the food is reported to cause immediate microbial inhibition, which is frequently followed by the recovery of injured cells [209]. Nowadays, natural antimicrobial compounds represent a valid alternative to chemical preservative agents, such as benzoic acid, propionic acid, sodium benzoate, sorbic acid, and potassium sorbate [16], for preserving food quality, because they can be effective against both food spoilage and foodborne pathogens [210], without constituting health concerns. The use of natural antimicrobials as preservative agents has, therefore, attracted increasing interest among consumers looking for clean food labels and more natural products. Several antimicrobial agents are present in nature, where they are produced mainly as secondary metabolites in microorganisms, plants, and animals, as defense mechanisms against exogenous threats. The incorporation of such natural compounds into edible coatings enables the development of active coatings, which combine physical protection of the food product (barrier effect) with significant antimicrobial activity.

Another class of compounds of interest for incorporation in ECs is represented by antioxidants. They are used to enhance the protection of fresh products and to increase their shelf-life as substances used to preserve food by retarding deterioration, rancidity, or discoloration due to oxidation caused by free radicals [211]. In this case, the synergy between the gas-barrier properties of the coating and the antioxidant activity is the key to a successful decrease in oxidation processes in coated foods [212].

ECs can be loaded also with anti-browning agents, contributing to reducing the extent of enzymatic and non-enzymatic oxidation of phenolic compounds during the shelf-life of fresh produce [213–215]. Anti-browning agents can be incorporated in cross-linking solutions and applied after the adhesion of the edible coating solution on the surface of fresh produce [216], for preservation during the entire storage period of food color, which is a critical quality parameter.

The organoleptic properties of the coated products can also be improved if the coating is loaded with flavoring or coloring agents, as well as with sweeteners, spices, and seasonings [5,18,178], which are also reported to provide health benefits.

Examples of NC addition in active systems are mainly reported in active films for food packaging, where the role of NC is of stabilization and physical entrapping of the active species. As shown in Table 8, the main effects of NC addition to films are related to (i) ensuring high loading of the antimicrobial agents [217] because of the intrinsic high surface area of NC, (ii) improving the controlled release characteristics of the bioactive agents loaded in the biopolymer matrix, by affecting their permeation rate [218], and, therefore, (iii) increasing the antioxidant properties of the film [219], when the payload bioactives are antimicrobial agents [7].

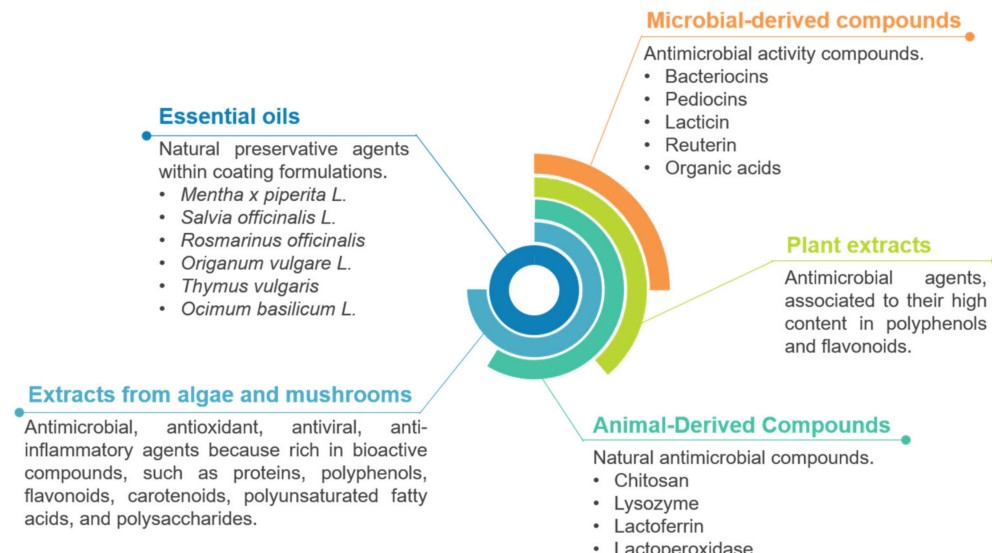

**Figure 2.** Natural compounds frequently used in edible active coatings.

**Table 8.** Recent advances in the antioxidant and antimicrobial properties of films reinforced with nanocellulose (literature data for years 2018–2021).

| Film-Forming Material | Additives | | Effect of NC on Active Film | References |
|---|---|---|---|---|
| | Reinforcing Agent | Active Agent | | |
| Sodium caseinate (4% *w/w*) | Cellulose nanofibers (2.5%–5% *w/w*) | Cinnamon bark essential oil-nanoemulsion (5% *w/w*) | NC decreases the release rate of the essential oil from sodium caseinate matrix and also improves the antioxidant properties of the film. | [218] |
| Soy protein (5% *w/v*) | Microfibrillated cellulose (0%–0.6% *w/v*) | Clove essential oil (2.5% *w/v*) | MFC's presence favors the release of the active compounds of CEO. A higher concentration of MFC increases the antioxidant properties as well as the antimicrobial activity. | [219] |
| Mucilage (50% *v/v*) | Cellulose nanofibers (3%–6% *w/v*) | - | NCs incorporation successfully enhances the mechanical, hydrophobic, antioxidant and antimicrobial properties of the mucilage composite films. | [217] |
| Gelatin/agar (2% *w/v*) | Cellulose nanofibers (0.75% *w/v*) | Clove essential oil-based Pickering emulsion (0, 0.02, 0.1, 0.2% *w/v*) | Composite film is transparent and shows high UV-light barrier properties and water-resistant properties, and improved antioxidant activity. | [220] |
| Poly (butylene adipate-co-terephthalate) (PBAT) (15% *w/w*) | Cellulose nanofibers (0.5, 1, 3% *w/w*) | Cinnamon essential oil | Films showed good thermal stability, higher oil release, decreasing water vapor permeability values and preventing microbial attack through the release of the essential oil. | [221] |

## 5. Conclusions

This review summarized the recent advancements about the incorporation of nanocellulose (NC) in a polymeric matrix to form edible coatings (ECs). Unlike NC used alone, which forms a coating with poor resistance to water vapor, the reinforcement of conventional coatings through the NC addition in the coating formulation is reported to significantly improve the ECs' properties. Remarkably, it was shown, through the critical analysis of the literature, how the properties of nanocomposite coatings, based on NC reinforcement, change depending on which type of NC (cellulose nanocrystals—CNC, or cellulose nanofibrils—CNF) and concentration are used. Therefore, the ECs' structural and chemical properties can be tailored through formulation, in combination with the selection of the optimal coating deposition technique.

Most of the studies to date have focused on the incorporation of CNC in coating solutions, because their high-crystallinity structure may increase the mechanical resistance (e.g., the tensile strength), as well as the barrier performance of the coatings. However,

CNC are reported to condense in the film-forming solutions, when their concentration is excessively high, with a consequent increase in water vapor permeability and a decrease in elongation at break value in the resulting coatings. In contrast, the use of CNF was reported to confer a more flexible structure to the coatings, due to their individual or aggregated softer and longer chains than CNC. The conflicting results reported, to date, about the effect of CNF on the properties of nanocomposite coatings, CNF has not been widely studied as a reinforcing agent. Bacterial nanocellulose (BNC) has recently emerged as a potential additive in ECs, because of its purity, which is reported to contribute to high tensile strength and mechanical flexibility to the coatings. The main limitation to the use of BNC currently resides in the production process of BNC-based composites, which needs to follow a bottom-up approach, with the need for bacterial growth for BNC production to take place in the presence of a matrix biopolymer. This restricts the possibility of changes in shape after fermentation, as well as generating high production costs.

Overall, the incorporation of NC in ECs represents a promising approach for improving ECs' mechanical and barrier properties, stability and eventual controlled release of active agents, with a potential impact in the preservation of the quality and extension of the shelf life of perishable fruits and vegetables with all-natural systems.

**Author Contributions:** Conceptualization, A.P. and F.D.; methodology, A.P.; investigation, A.P.; resources, G.F.; writing—original draft preparation, A.P.; writing—review and editing, G.F. and F.D.; supervision, F.D.; funding acquisition, G.F. All authors have read and agreed to the published version of the manuscript.

**Funding:** This research was funded by the Italian Ministry of University (MUR) call PRIN 2017 with the project 2017LEPH3M "PANACEA: A technology PlAtform for the sustainable recovery and advanced use of NAnostructured CEllulose from Agro-food residues".

**Institutional Review Board Statement:** Not applicable.

**Informed Consent Statement:** Not applicable.

**Conflicts of Interest:** The authors declare no conflict of interest.

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
