# Peer review of "The Use of Nanocellulose in Edible Coatings for the Preservation of Perishable Fruits and Vegetables"

_coatings, doi:10.3390/coatings11080990_

Round 1

Reviewer 1 Report

Figure 1c shows actually spreading (in text 1d) and 1d shows dipping (in text 1c). This should be corrected, i.e. switch placed in figure.

line 133: continue the sentence "...the surface of the substrate."

The sentence on line 134 "The excess dilutent leaves the coating solution thin film on the food surface", please rewrite this sentence as it is not understandable in this form.

line 274: the linkages are glucosidic linkages since only glucose units are found

Author Response

Responses to Reviewer #1

Figure 1c shows actually spreading (in text 1d) and 1d shows dipping (in text 1c). This should be corrected, i.e. switch placed in figure.

R: We thank the reviewer for pointing out this mistake. We have now switched the figure of spreading and dipping technique.

line 133: continue the sentence "...the surface of the substrate."

R: We have now added the text to the sentence.

The sentence on line 134 "The excess dilutent leaves the coating solution thin film on the food surface", please rewrite this sentence as it is not understandable in this form.

R: As suggested by the reviewer, we have modified the text, which now reads: “The excess diluent leaves the food surface by evaporation at room temperature or drying with heated air, thus achieving a thin film of the coating solution.”.

line 274: the linkages are glucosidic linkages since only glucose units are found

R: We have modified the text ss suggested by the reviewer. The manuscript now reads (lines 284-285): “Cellulose exhibits a unique molecular structure, consisting of a linear homopolysaccharide composed of glucose monomers, linked together by β-1-4-glycosidic bonds,…”

Reviewer 2 Report

It has been a great honor, as well as a pleasantly challenging activity, to review the article entitled “The use of Nanocellulose in Edible Coatings for the Preservation of Perishable Fruits and Vegetables” by Annachiara Pirozzi et al.

The article is structured following the classic model for the Review Article. The major components of the article are presented coherently and logically, tightly linked to one another. The list of bibliographic references is adequate, the documentation is appropriate regarding the titles consulted. The paper is of high value, and it treats a specific subject that is of high interest in the domain of the edible coatings and in the use of nanocellulose.

Above the work presented by the authors be very good, I think that the organoleptic aspect of these coatings, over the food, must be discussed. Another aspect that I think that should be addressed, is the fact that calling these coverings "edibles" is no consentaneous within the scientific class. So, in this way, the authors must clarify, in this work, if the films are edible by the consumer with the food "covered". If so, a small approach should be made about how these foods will be available on the market. They will have a secondary packaging protecting them or not?

I recomend the attention for the  the following comments.

_Line 99-100 – Replace “treatment o reduce” by “treatment to reduce”.

_Line 103 - Authors should further explain what the terms "contour coating” and “contactless coating" refers to.

_Line 306 – Table 3 - It was important that the authors add the dates of the different references in the table.

_Line 340 – Table 5 - It was important that the authors add the dates of the different references in the table.

_Line 481-494 - Missing reference.

_Line 496-510 - Missing reference.

_Line 574 – Table 8 - It was important that the authors add the dates of the different references in the table.

Author Response

Responses to Reviewer #2

It has been a great honor, as well as a pleasantly challenging activity, to review the article entitled “The use of Nanocellulose in Edible Coatings for the Preservation of Perishable Fruits and Vegetables” by Annachiara Pirozzi et al.

The article is structured following the classic model for the Review Article. The major components of the article are presented coherently and logically, tightly linked to one another. The list of bibliographic references is adequate, the documentation is appropriate regarding the titles consulted. The paper is of high value, and it treats a specific subject that is of high interest in the domain of the edible coatings and in the use of nanocellulose.

R: We sincerely thank the reviewer for appreciating our work and for the positive comments.

Above the work presented by the authors be very good, I think that the organoleptic aspect of these coatings, over the food, must be discussed. Another aspect that I think that should be addressed, is the fact that calling these coverings "edibles" is no ithin the scientific class. So, in this way, the authors must clarify, in this work, if the films are edible by the consumer with the food "covered". If so, a small approach should be made about how these foods will be available on the market. They will have a secondary packaging protecting them or not?

R: We fully agree with the reviewer, and we thank you for the insightful comment. We have now modified the introduction by adding a brief paragraph, which clarifies the above mentioned aspects (lines 35-43): “ECs acts as primary (closest to food) packaging. Thus, the main advantage over traditional synthetic packaging is that ECs can be consumed with the food, with no package to dispose of [4], reducing the cost and complexity of packaging systems designed to protect fresh perishable foods. Moreover, also if they are not eaten by the consumers, ECs could still contribute to the reduction of environmental pollution, because they are produced exclusively from renewable, edible ingredients and therefore degrade more readily than polymeric materials [5]. In addition, because of the additional protection they offer, ECs enable also the simplification of the secondary packaging (next layer of packaging), making recycling more accessible [6].”.

I recomend the attention for the following comments.

_Line 99-100 – Replace “treatment o reduce” by “treatment to reduce”.

R: We have now modified the text.

_Line 103 - Authors should further explain what the terms "contour coating” and “contactless coating" refers to.

R: As suggested by the reviewer, we have now added explanation of "contour coating” and “contactless coating". Now the text reads (lines 111-114):“such as contouring or contour coating (the coating solution freely falls across the sub-strate and forms a curtain, which follows the substrate contours) and contactless coating (no mechanical contact takes place between the base substrate and the distributor of the coating solution)…”.

_Line 306 – Table 3 - It was important that the authors add the dates of the different references in the table.

R: Actually, the references cited in the Table 3 are only from years 2020 and 2021. This has now been specified in the table legend, which reads: “Recent Advances in the Production Process, Physicochemical Characteristics and Application of CNC Extracted from Different Sources (Literature data for years 2020 and 2021)”

_Line 340 – Table 5 - It was important that the authors add the dates of the different references in the table.

R: Also the references cited in the Table 5 are only from years 2020 and 2021. This has now been specified in the table legend. Moreover, we did the same for Table 4.

_Line 481-494 - Missing reference.

R: We agree with the reviewer and we have now added the following reference:

  1. Koh, P.C.; Noranizan, M.A.; Karim, R.; Nur Hanani, Z.A.; Lasik-KurdyÅ›, M. Combination of alginate coating and repetitive pulsed light for shelf life extension of fresh-cut cantaloupe (Cucumis melo L. reticulatus cv. Glamour). J. Food Process. Preserv. 2018, 42, 1–18, doi:10.1111/jfpp.13786.

_Line 496-510 - Missing reference.

R: We have now added the following references to the one previously cited:

  1. Abu Salha, B.; Gedanken, A. Extending the Shelf Life of Strawberries by the Sonochemical Coating of their Surface with Nanoparticles of an Edible Anti-Bacterial Compound. Appl. Nano 2021, doi:10.3390/applnano2010002.
  2. Sharif, Z.I.M.; Jai, J.; Subuki, I.; Zaki, N.A.M.; Mustapha, F.A.; Mohd Yusof, N.; Idris, S.A. Optimization of Starch Composite Edible Coating Formulation on Fresh-Cut “fuji” Apple through Surface Tension, Wettability and FTIR Spectroscopy. In Proceedings of the IOP Conference Series: Materials Science and Engineering; 2019.
  3. Rai, S.; Suman Rai, C.; Poonia, A. Formulation and characterization of edible films from pea starch and casein. J. Pharmacogn. Phytochem. 2019.
  4. Leyva-Porras, C.; Cruz-Alcantar, P.; Espinosa-Solís, V.; Martínez-Guerra, E.; Piñón-Balderrama, C.I.; Martínez, I.C.; Saavedra-Leos, M.Z. Application of differential scanning calorimetry (DSC) and modulated differential scanning calorimetry (MDSC) in food and drug industries. Polymers (Basel). 2020.

_Line 574 – Table 8 - It was important that the authors add the dates of the different references in the table.

R: The references cited in the Table 8 are from the years 2018-2021. This has now been specified in the table legend, which reads: “Recent Advances on the Antioxidant and Antimicrobial Properties of Films Reinforced with Nanocellulose (Literature data for years 2018-2021).”

Round 2

Reviewer 1 Report

The corrections look fine and the manuscript can be accepted in this form.